# The Development of Algorithms for Individual Ranges of Body Temperature and Oxygen Saturation in Healthy and Frail Individuals

**DOI:** 10.3390/healthcare12232393

**Published:** 2024-11-28

**Authors:** Märta Sund Levander, Ewa Grodzinsky

**Affiliations:** 1Department of Health, Medicine and Caring Sciences, Medical Faculty, Linköping University, 58185 Linköping, Sweden; 2Department of Biomedical and Clinical Sciences, Medical Faculty, Linköping University, 58185 Linköping, Sweden; ewagrodzinsky@gmail.com

**Keywords:** algorithms, body temperature, habitual condition, individual normal, interpretation, oxygen saturation, precision medicine

## Abstract

Background/Objectives: Individual habitual conditions entail a risk during the interpretation of vital parameters. We developed algorithms for calculating, validating, and interpreting individual normal ranges of body temperature and oxygen saturation. Methods: In total, 70 healthy individuals aged 27 to 80 and 52 frail individuals aged 60 to 100 were included. Data on individual conditions comprised age, gender, physical ability, chronic disease, and medication. Ear temperature and oxygen saturation were measured for five mornings before the participants got out of bed and consumed medicine, food, or drink. Results: The range for body temperature was 34.3 °C to 37.7 °C, with a variation of 0.7 °C ± 0.4 °C. The variation in minimum and maximum temperatures was 2.4 °C vs. 2.7 °C and 2.9 °C vs. 2.3 °C in healthy and frail subjects, respectively. The range for oxygen saturation was 85% to 99% in healthy individuals and 75% to 100% in frail individuals. The variation between minimum and maximum oxygen saturation was 13% vs. 25% and 4% vs. 17% in healthy and frail subjects, respectively. Conclusions: To promote the implementation of precision medicine in clinical practice, it is necessary to interpret body temperature and oxygen saturation based on individual habitual conditions. Interpreting deviations from an individual’s normal ranges allows healthcare professionals to provide necessary treatment without delay, which can be decisive in preventing further deterioration.

## 1. Introduction

Precision medicine, also called individual-based, person-based, or tailored medicine, aims to provide treatment according to a patient’s unique biological variables, conditions, and needs. The potential of precision medicine lies in its ability to guide clinical decisions toward the most effective treatment for a given patient, thus improving the quality of care while reducing the need for unnecessary diagnostic tests and therapies [1,2,3]. Although no official consensus exists on this approach [3], the focus is on the early diagnosis of genetically caused diseases through screening methods and tailored treatment. However, the reference intervals used for the assessment of vital parameters and biochemical markers are still mainly derived from healthy individuals. People with long-term medical conditions and daily medication regimens are not included in reference populations. The limits considered for younger adults are not always applicable because of age-associated physiological changes and increased inter-individual differences with age [4]. Additionally, despite the increasing personalization of medicine, 37.0 °C persists as the estimate of normal temperature [5] and is a legitimate reason for accessing the healthcare resources available for patients seeking care [6,7]. To ensure patient safety and measurement accuracy, healthcare professionals use standardized clinical decision support tools to assess and manage a patient’s deteriorating health condition [8,9].

The conventional definitions of normal body temperature as 37 °C and a fever state as ≥38 °C are based on measurements made by the German physician Wunderlich from the middle of the 19th century [10,11]. However, Wunderlich made these measurements by placing a primitive thermometer under the armpit of sick, but not healthy, individuals [6]. Since then, research has revealed a large variation between groups of individuals and that the normal body temperature is lower than its conventional definition [5,6,12,13,14,15,16].

When changing health conditions are assessed, frail elderly adults are particularly vulnerable because they often suffer from multiple chronic diseases and present nonspecific signs and symptoms, including the absence of a temperature of >38 °C, in the presence of infectious diseases [17,18,19,20,21,22]. Deteriorated health conditions due to chronic diseases and aging lead to impaired regulatory systems, resulting in low-grade chronic inflammation [23], which has been found to affect normal ranges of common biochemical markers [24,25]. Clinical consequences include delayed, incorrect, or a lack of diagnosis and treatment [17,19,20,21,26,27]; an increased risk of severe illness and death; the need for hospitalization; long recovery; and persistent disability. In 2021, the care cost per day in somatic care at county hospitals in Sweden was approximately SEK 13,000 [28]. Unnecessary transport and hospitalization lead to suffering and changes in the environment for individuals and healthcare costs for society [29]. Previous research revealed that 40% of NHRs admitted to a hospital could have received safe treatment at the facility [30].

The challenge of defining a “normal” body temperature became evident when screening for COVID-19 during the pandemic. Individuals positive for COVID-19 but with mild symptoms were assessed as healthy because they did not have a fever defined as a temperature > 38 °C [31]. Lippi et al. [32] also emphasized uncertainty in temperature measurement procedures. During the pandemic, it also became clear that deterioration in oxygen saturation was crucial for detecting ongoing infection. As with body temperature, the lowest value for normal oxygen saturation varies, with studies reporting 90% [33], 91% [4], 95% [34], 94% [35], 96% [36], and 97% [37]. Möckel et al. [38] reported oxygen saturation < 95%, together with dyspnoea and a history of cardiovascular problems, as the important variables for distinguishing between high- and low-risk groups of COVID-19-positive patients. However, during the pandemic, patients were admitted to hospitals with “silent hypoxia”, i.e., oxygen saturation <90% and no dyspnea, despite severe infection [39]. Lehnertz et al. [40] found at least one arterial oxygen saturation (SpO_2_) value of ≤94% 14 days prior to symptom onset in 70% of elderly nursing home residents (NHRs) and suggested a mean decrease of at least 3% between repeated measurements to be an indicator of an acute change in detecting suspected COVID-19 infection.

Since normal body temperature varies between individuals [5,6,41,42,43], it is self-evident that the same also applies to temperature assessed as corresponding to a fever [5,6,41,44,45]. In a large multicenter study, we measured ear-based body temperature in seemingly healthy individuals and asked them to self-report the individual thresholds they considered fever temperatures. Based on the results, we suggested at least a 1 °C increase from an individual’s normal temperature, DiffTemp^®^, together with malaise, to be a more accurate definition of fever temperature [12]. When DiffTemp^®^ was applied to NHRs, 49/49 frail elderly patients with verified infection were assessed as being febrile, which can be compared to 16/49 (33%) when ≥38 °C was used as the benchmark [44].

In another study involving patients with acute brain injury in a neurosurgical intensive care unit (NICU), DiffTemp^®^, i.e., the temperature difference from the morning temperature on the day before, exhibited an odds ratio [OR] of 17.4 for confirmed infection as compared to suspected infection, while ≥38 °C was not associated with confirmed infection. Additionally, diagnostic sensitivity analysis revealed that DiffTemp^®^ had 90.9% specificity for patients with no infection and 93.8% sensitivity for patients with confirmed infection [46].

Regarding normal variation in oxygen saturation, Vold [36] reported a 6.3% prevalence of SpO_2_ ≤ 95% in the general population. Those over 70 years of age may have oxygen levels closer to 95%, which can be normal for this age group [47]. The ability to adjust oxygen saturation in order to adapt to different conditions such as high altitude and changes in barometric pressure varies between individuals [48,49]. Chronic pulmonary disease and obesity also alter oxygen saturation [36]. Lapin et al. [47] emphasized that one’s oxygen saturation level varies considerably based on their state of health. Thus, it is important to understand both baseline readings and underlying physiology associated with certain conditions to interpret oxygen saturation levels and changes in these levels. Measurement uncertainty regarding finger pulse oximeters has also been reported, as they may register values that are too high when considering oxygenation in arterial blood, indicating failure in detecting hypoxia [50]. Measurement accuracy is also affected by poor circulation, dark skin, subcutaneous tissue, tobacco use, and nail polish when oxygen saturation is measured with a finger pulse oximeter [36,51].

Larjani et al. [3] emphasized that a more individualized approach is needed because of significant individual variations. We argue that to fully apply precision medicine clinically, changes in vital parameters, measured using standardized methods, must be interpreted based on an individual’s normal range in habitual conditions. If the interpretation is based on an unreliable practice, it will further contribute to uncertainty in precision medicine [52]. We argue that as long as fever and oxygen saturation are defined based on tradition and not science, we will not achieve accuracy in precision medicine. Thus, this study aimed to develop algorithms for the calculation and interpretation of individual normal ranges for body temperature and oxygen saturation.

## 2. Materials and Methods

This study was a cohort study with a descriptive and prospective design.

### 2.1. Participants

The sample size was calculated based on the international standard for the clinical testing of thermometers [53]. The group comprising middle-aged healthy individuals was designated as “healthy”, and “frail elderly” included nursing home residents (NHRs) and patients undergoing hospital-affiliated home care (LAH).

#### 2.1.1. Healthy Subjects

A convenience sample of healthy individuals from the authors’ circle of acquaintances participated in this study. The healthy people managed their daily lives independently and without the need for any support from society, lived in their own accommodations, and worked professionally or were retired. Although the healthy participants’ habitual conditions could include chronic illness managed with prescription drugs, they were considered healthy because they were all living in their own homes with no need for support for managing their daily lives. There were no exclusion criteria for healthy individuals.

#### 2.1.2. Frail Subjects

Frail subjects included NHRs ≥ 65 years of age from nursing homes and patients ≥ 60 years of age undergoing LAH. They were all frail, considering a combination of aging, comorbidity, physical and/or cognitive decline, risk of malnutrition, and need for around-the-clock care and assistance in daily-life management [54]. The frail elderly people lived in three non-profit municipality nursing homes and received support to help them cope with daily life around the clock. Patients who underwent LAH were cared for either in their own homes with around-the-clock care or in a special hospital ward. People receiving terminal care were not invited to participate.

### 2.2. Data Collection

#### 2.2.1. Background Data

Descriptive data (age and gender) and data on habitual conditions, including physical ability, chronic diseases (such as cardiovascular disease, hypertension, kidney disease, autoimmune disease, dementia, stroke, and malignancy), and medication, were obtained directly from the acquaintances and from nursing homes and patient records and compiled in coded protocols. Documentation about laboratory and genetic markers was not registered in this study.

#### 2.2.2. Physical Activity

Physical activity, i.e., the ability to perform activities in daily life, was assessed using Activities of Daily Living (ADL) [55]. The total cumulative score involves the assessment of 10 categories reflecting personal (food intake, continence, transfer, going to the toilet, dressing/undressing, and bathing) and instrumental (cooking, transport, food-shopping, and cleaning) abilities. The ratings range from 0 to 10, where 0 = independent and 10 = dependent in regard to all 10 activities. Grade 5 means that a person can handle food intake, continence, movement, going to the toilet, and dressing/undressing by themselves.

#### 2.2.3. Measurement of Body Temperature and Oxygen Saturation

Ear-based body temperature was measured using infrared technology; healthy subjects used Genius 2 (Medtronic, Solna, Sweden) to take measurements, and for NHRs and LAH patients, Thermoscan (6 IRT6515, Braun, Kronberg im Taunus, Germany) was used. Oxygen saturation (SpO_2_) was measured via fingertips with a Beurer pulse oximeter (Beurer, Sundbyberg, Sweden). The Genius 2 and Thermoscan thermometers were set to measure the actual temperature without predetermined settings for adjustment to another measuring site. Genius 2 was calibrated by the Department of Medical Technology University Hospital in Linköping. The Thermoscan thermometers and Beurer pulse oximeters were calibrated before use, according to the manufacturers’ standards, for quality assurance. Both measurements were performed in the morning for 5 days using a standardized method, i.e., before getting out of bed and consuming medicine, food, or drink.

#### 2.2.4. Assessment of Normal/Habitual Stat

The Early Detection of Infection Scale (EDIS) was used as an aid for the nursing staff for addressing NHRs and LAH patients if they suspected that the persons were not in their habitual state. EDIS involves the assessment of changes from habitual conditions with suspected changes in health status. The EDIS comprises more general behavioral changes, expressed as “he/she is not as usual” and discomfort, such as facial expressions, confusion, aggressiveness, weakness/apathy, unrestrained, restlessness, and altered food intake, as well as more obvious signs and symptoms of infection, expressed as “he/she seems to be ill”, such as fever, chills, or pallor [21,22,44].

### 2.3. Performance

#### 2.3.1. Healthy Subjects

After providing written informed consent, participants responded to a questionnaire about chronic disease and medication and the presence of malaise and or having undergone vaccination in the last three days. If no symptoms of infection were present, they measured their body temperature in both ears and oxygen saturation via a finger for five days before getting out of bed and consuming medicine, food, or drink. The subjects themselves documented measurements in coded protocols.

#### 2.3.2. Frail Subjects

After we obtained informed consent from the participants or via consultation with their next of kin and when each individual was in habitual condition, body temperature in both ears and oxygen saturation were measured for five days before the patients got out of bed and consumed medicine, food, or drink. In the nursing home, the nursing staff performed all measurements but measured the right-ear temperature only. Regarding the LAH patients, some underwent home care, while others were admitted to a hospital. Hence, some patients undergoing home care performed the measurements themselves after receiving instructions from the nursing staff, while the nursing staff performed all measurements for inpatients.

Registered nurses at the nursing homes and LAH facility compiled background data on NHRs and LAH patients from the medical and nursing records.

### 2.4. Considerations in Algorithm Development

To reduce the influence of diurnal variation, activities, and metabolism, measurements were carried out using a standardized approach at the same time 5 mornings before the patients rose from bed and consumed food, drink, or medicine. We used the same site (ear) due to temperature gradients within the body. The ear site is close to the hypothalamus.

The development of the algorithms roughly involved the determination of technical measurement uncertainty according to international standards (ISO) [53,56] and physiological and individual variations [6,12], indicating the reproducibility of repeated measurements, including the 95% CI of the mean differences between measurements. Based on our previous large multicenter study [12], too large a deviation when establishing the individual normal range may indicate ongoing immunological processes affecting body temperature. We also studied inter-individual variability but found no statistically significant difference between repeated measurements performed by two operators [57]. The algorithms approved the measured values and established an individual reference interval (IRef^®^). If the algorithms did not approve a value after 5 days of measurement, the measurement continued for 1–2 more days, and the algorithms recalculated the value. If the measurement data still did not confirm the results, the procedure was postponed for 7–10 days and then repeated from the beginning, as deviant values may relate to procedure errors or an ongoing infection, indicating that the reference interval cannot be established in an individual’s habitual condition.

### 2.5. Statistical Analysis

Data were analyzed using descriptive statistics (PASW Statistics 29 software, SPSS Inc., Chicago, IL, USA). The level of significance was set to *p* < 0.05. Student’s two-sided *t*-test was used to compare mean values between groups, and Pearson’s Rho correlations between variables were included in the regression analysis. Binary logistic regression analysis was used to analyze conditions related to body temperature and oxygen saturation [58]. A low body temperature was defined as a max temperature of 36.2 °C or below, and a higher body temperature value was designated as 36.3 °C or above, based on the 10th percentile as a lower limit for ear temperature in frail elderly persons [59]. According to the WHO, the world’s leading organization for health guidelines, low oxygen saturation was defined as 94% or below, and high levels were considered to be 95% or above [35]. In total, we performed 119 and 99 measurements in the right ear and left ear, respectively. As there were missing data for left-ear temperature, and no difference was observed between simultaneous measurements between the right and left ears (36.2 °C ± 0.6 °C for both ears, irrespective of the day of measurement), we used the values from the right-ear measurement for statistical analysis. As ADL, dementia, and nutrition correlated to a greater degree than Pearson’s Rho 0.4, these variables were analyzed in a separate regression analysis (see Table 1).

## 3. Results

The final sample consisted of 70 healthy individuals (46 of whom were women), with a mean age of 61 ± 13 years (ranging from 27 to 80, with 2 patients with missing data); 52 frail elderly individuals, comprising 20 NHRs (13 women), with a mean age of 87 ± 8 years (ranging from 70 to 100); and 32 patients undergoing LAH (10 women), with a mean age of 79 ± 8 years (ranging from 60 to 95). Among healthy subjects, 59/70 (84%) had no (*n* = 33) or one (*n* = 26) diagnosis, mainly hypertension, while the majority of frail individuals (34/52; 65%) had more than two diagnoses. Frail elderly subjects more frequently had multiple illnesses and took more medication daily. Among frail participants, all but five LAH patients scored high (7 ± 2 in NHRs and 4 ± 3 in LAH patients), which can be compared to the value of 0 for healthy individuals. Frail individuals were also at risk of malnutrition, whereas none of the healthy individuals exhibited this risk (see Table 2).

No differences were observed in the variation in the normal range of body temperatures between healthy and frail individuals (0.7 °C ± 0.4 °C for both groups), while the normal range in oxygen saturation was greater for frail elderly than healthy subjects (5% ± 3% vs. 3% ± 2%, *p* < 0.001). No differences were found between male and female participants, neither in terms of body temperature nor oxygen saturation. The total body temperature ranged from 34.3 °C to 37.7 °C and 34.5 °C to 37.7 °C in healthy and frail individuals, respectively. The mean right-ear morning body temperature was lower in the healthy subjects than in the frail elderly patients (36.1 °C ± 0.4 °C vs. 36.6 °C ± 0.5 °C, *p* < 0.001). A maximum temperature of 36.2 °C or below was more frequent among healthy subjects than among frail elderly patients (26/68 (38%) and 3/51 (6%), respectively). We found variations of 2.4 °C and 2.7 °C in minimum temperature values and 2.9 °C and 2.3 °C in maximum temperature values among healthy subjects and frail elderly participants, respectively (see Table 3 and Figure 1).

The risk of malnutrition or heart disease, dependency on daily functioning, and a combination of conditions related to frailty were negatively related to a lower body temperature and positively related to a higher body temperature (see Table 4). The presence of COPD, asthma, dementia, or diabetes was associated with maximum oxygen saturation of 94% or below, while COPD, asthma, dementia, and heart disease were negatively associated with an oxygen saturation level of 95% or above (see Table 5).

When including medication in the logistic regression, painkillers and sedatives were negatively associated with a maximum body temperature of 36.2 °C or below (Exp B: 0.250, 95% CI: 0.093 to 0.671, and *p* < 0.01; Exp B: 0.173, 95% CI: 0.038 to 0.779, and *p* < 0.05, respectively). The same drugs were positively associated with a temperature of 36.3 °C or above (for painkillers, Exp B: 4.008, 95% CI: 1.491 to 10.774, and *p* < 0.01; for sedatives, Exp B: 5.786, 95% CI: 1.284 to 26.070, and *p* < 0.05). Antidepressants were positively related to oxygen saturation of 94% or below (Exp B: 11.418; 95% CI: 1.636 to 79.673; *p* < 0.01) and negatively to oxygen saturation of 95% or above (Exp B: 0.088; 95% CI: 0.013 to 0.611; *p* < 0.014).

## 4. Discussion

To the best of our knowledge, this is the first study presenting body temperature and oxygen saturation based on individual normal ranges taking into account habitual conditions in terms of chronic disease and medication. Recently, Ley et al. [60] presented the normal range in oral temperature based on machine learning algorithms; however, the normal range was based on mean values determined at the group level.

In the present study, we focused on the interpretation of body temperature and oxygen saturation, mainly because we have investigated body temperature measurement for several years and gained insight into the significance of deviations in oxygen saturation during the COVID-19 pandemic. Assessing the health status of frail elderly adults is a clinical challenge due to the complexity of aging, frailty, and atypical presentations [17,18,19,20,21,22]. In frail individuals, the risk of malnutrition; impaired physical and cognitive abilities; and daily medication with analgesics, sedatives, and antidepressants were associated with a body temperature of 36.3 °C or above. Over a 10-year period, it was found that people admitted to nursing homes in Sweden had become older and more frail, and their survival time had decreased [61]. Since that study, another 10 years have passed, so it is reasonable to assume that, currently, NHRs are even more fragile. Our findings indicate that frail elderly adults are in special need of assessment of health conditions based on individual normal values.

Others have also reported a large variation in mean body temperature [13,14,15,62,63]. Even though the mean and SD values differed between healthy and frail elderly subjects in the present study, the variation in body temperature, measured five days in a row, was similar between groups, while the range for oxygen saturation demonstrated a larger variation in the frail elderly participants. The variations in minimum and maximum values (approximately 3 °C for body temperature and 13% for oxygen saturation) in both the healthy and frail individuals further highlight the broad range of normal body temperature and oxygen saturation.

Contradicting the present results, earlier research reported a lower body temperature in frail elderly adults [59,64]. Research has also revealed that impaired immune defense in the aging body is associated with low-grade nonspecific inflammation, i.e., an increased concentration of pro-inflammatory cytokines and acute-phase proteins, linked to chronic conditions, such as COPD, cardiovascular disease, and dementia [4]. This might explain the higher mean body temperature in the frail elderly participants in the present study.

However, it is not as obvious whether the variation in oxygen saturation differs from current reference values. No healthy subject had a maximum oxygen saturation of 94%, whereas it was more common in frail elderly individuals. Aside from pulmonary disease, this may be attributed to the presence of heart disease, which, in frail elderly adults, is associated with oxidative stress and chronic inflammation, impairing oxygen saturation [65,66]. The results of this study, although corresponding to lower values, are in line with the findings of Rodríguez-Mumoli et al. [4], who reported that the normal limit for SpO_2_ in elderly individuals without pulmonary disease was 91%, similar to that of the younger population.

Assessing changes in health status based on an individual’s unique conditions has serious health–economic consequences. Currently, unnecessary hospitalizations and hospital admissions of NHRs generate excessive healthcare expenditures for society [29]. Previous research has revealed that 40% of NHRs admitted to a hospital could have received safe treatment at the facility [30]. Ouslander et al. [29] reported that there was a reduction in hospitalizations when care models engaged advanced practice nurses and involved collaboration with physicians. Improving the management of acute changes in health conditions reduced hospitalization in nursing homes in the US by 17% [67]. Care models and decision support can thus prevent unnecessary hospitalizations; however, we argue that as long as the assessment is not based on individuals’ conditions, it is not enough. The application of precision medicine with individualized assessment would enhance earlier diagnosis and further reduce hospitalization. In clinical practice, rapid response systems are designed to detect changes in vital parameters early [66], for instance, using the RETTS (rapid emergency triage and treatment system). The RETTS is used to assess and manage deterioration in acute illness [68]. The NEWS2 (National Early Warning Score 2) [69] quantifies the severity of deterioration by scoring each parameter with respect to reference values. However, age-specific normal limits for several vital signs and physiological parameters have not yet been established for the elderly population [4], posing a risk of misdiagnosis and consequences for both the individual and the healthcare system.

The concept of personalized medicine was already put forward in 1944 by Iwy [70], who emphasized the significance of understanding variability and carefully defining normal or abnormal levels to avoid mistakes in the interpretation of an individual’s condition. Galen and Gambino [71] further stated that the concept of normality is itself inadequate for the proper interpretation of test results if it is not considered in association with a reference value. They revealed that meaningful reference values can only be obtained by studying and defining reference populations. However, elderly individuals constitute a heterogeneous group ranging from individuals with diseases and impairments to healthy individuals managing their daily lives independently. In addition, changes in biomarker levels are unclear since these changes could be affected by aging per se or disease.

Stryuf et al. [72] underlined the importance of understanding the accuracy of diagnostic tests in order to identify effective diagnostic and management pathways. The technical measurement uncertainty in the instruments used clinically to measure body temperature and oxygen saturation is well regulated by quality assurance systems and international standards [53,56]. However, the clinical measurement uncertainty, i.e., how reference and limit values are measured, calculated, and interpreted, must be updated based on research and evidence. Hence, it is not surprising that, as declared by Lohse [73], uncertainty seems to be a key characteristic of precision medicine in practice, in particular regarding clinical decision-making. We argue that even if the technical quality and measurement procedure used meet requirements for accuracy, the basis for how the reference and cut-off values are calculated can still jeopardize interpretation in clinical practice. To promote clinical precision medicine, a different approach should be adopted, one that bases the assessment of health status on an individual’s normal range in habitual conditions and not group-level cut-offs.

We believe that it is time to implement and validate the concept of IRef^®^ and deviations from IRef^®^, i.e., DiffTemp^®^ and DiffOx^®^, to apply both precision medicine and evidence-based care in clinical settings when assessing a changed health state. Using an individual’s normal range to interpret deviations enhances the possibility of providing necessary care without delay, which can be decisive before further deterioration and in preventing inaccurate treatment and hospitalization. Concerning body temperature, this is supported when using a ≥1 °C difference from an individual’s normal temperature to asses fever among NHRs [21,44] and critically ill patients in NICUs [46], but for oxygen saturation, more data are needed to explore critical deviations from individual normal ranges.

## 5. Clinical Implications

In terms of precision medicine, our approach is a time- and cost-saving genetic-free solution for early filtration in the reception of people for hospitalization vs. home-care supervision. To implement assessment based on an individual’s normal range, we need to use different approaches. We can already document an individual’s normal range in a habitual state by measuring body temperature and oxygen saturation five mornings before they rise and consume food, drink, or medication and then using the difference (Difftemp^®^ and Diffox^®^) when assessing deteriorating health. Decisive for a pervasive, sustainable change in clinical routines is the fact that knowledge management and documentation systems adopt and recommend the new approach. In future research, we plan to implement the algorithms in existing measuring instruments for body temperature and oxygen saturation to accommodate measurement uncertainty, including sensitivity analysis, thus facilitating the use of these tools by both professionals and ordinary people. These results will be the basis for the continued development and clinical testing of the software platform with our algorithms.

As a concluding remark, we present a real-life scenario of a frail elderly nursing home resident with pneumonia. Asta, 91 years old, lives in a nursing home. She has dementia, heart disease, and autoimmune disease; she is malnourished and depressed; and her body aches. She receives T Paracetamol (1 g) 3–4 times per day. Her body temperature measured 3 days in the morning before medication, food, and drink consumption was 35.7 °C in her right ear. One morning, Asta was lying on the floor. The next day, she had a temperature of 37.2 °C measured via her ear, and she was not as usual. After 4 days, Asta received antibiotics for pneumonia but deteriorated on day 5, with a temperature of 38.4 °C, and was transferred to the hospital by ambulance. After 8 days in the hospital, Asta died. This scenario illustrates that, on day 2 of temperature assessment, had the definition of fever been based on the individual’s normal range, it would have led to early diagnosis and treatment and may have also avoided transfer to the hospital. Her ear temperature then was 37.2 °C, i.e., Difftemp^®^ +1.5 °C. In the case of Asta, the costs for transport to the hospital and inpatient care reached SEK 120,000 [28] (see Table 6).

The strength of our study is that we have refined precision medicine by developing algorithms that calculate and interpret an individual’s normal ranges for body temperature and oxygen saturation based on research as well as technical and physiological variation. We performed measurements using a standardized approach and chose a measuring site close to the hypothalamus and temperature-sensitive neurons, i.e., the ear. We considered inter-individual variability between measurements and technical device accuracy. A limitation is that we did not confirm illness with notes from medical records for healthy subjects, and we did not document lifestyle factors, e.g., sleep habits. Reported daily medication, though, helped to confirm self-reported diseases. There is also always a risk that people carry diseases they are not aware of. Another limitation involves the small sample of frail elderly participants. Frail elderly people are often excluded in studies related to the assessment of health status, although they constitute the largest group in need of medical care. Their inclusion is thus a strength of our study. Lastly, a limitation is that we have studied body temperature over several years, but we do not have the same experience regarding variation in oxygen saturation.

## 6. Conclusions

The present results confirm there are large variations in body temperature and oxygen saturation among healthy and frail individuals. The results indicate that the general reference values currently used at the group level do not match the degree of variation in body temperature and oxygen saturation in practice. We argue that our results promote the realization of using precision medicine in clinical practice and can easily and directly be applied. Measurements taken to determine the normal range should be performed in a standardized approach when an individual is in a habitual condition. To facilitate the measurement procedure, we have developed algorithms to automatically measure, calculate, and approve inter-variability between measurement values.

The interpretation of deviations using an individual’s normal range increases the possibility of providing necessary care without delay, which can be decisive in preventing further deterioration, inaccurate treatment, and hospitalization, especially among frail elderly individuals.

## Figures and Tables

**Figure 1 healthcare-12-02393-f001:**
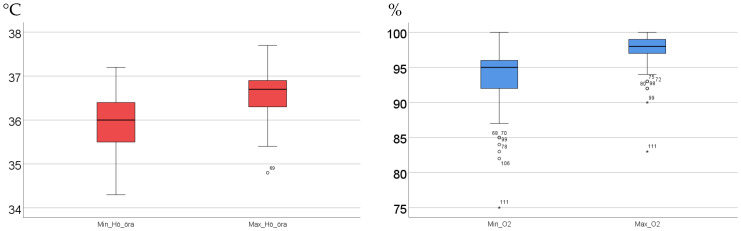
Variation in minimum and maximum values for body temperature (°C) and oxygen saturation (%), measured 5 mornings in a row, among 70 healthy and 52 frail elderly subjects. The graph illustrates the median value (bold line) and variation between the 25th and 75th percentiles within the box. Unfilled circles represent values between 1.5 and 3 times the interquartile range. The asterisks represent values that are more than 3 times the interquartile range. The numbers correspond to individuals.

**Table 1 healthcare-12-02393-t001:** Significant correlations (Pearson’s Rho) between variables included in the regression analysis.

Condition	Dementia	Heart Disease ^a^	Asthma	COPD ^b^	Stroke	Diabetes	Liver Disease	Kidney Disease	Autoimmune Disease	ADL ^c^	Nutrition
Dementia		0.290 **			0.217 *	0.240 **				0.581 **	0.450 **
Heart disease ^a^	0.290 **		0.023	0.216 *	0.215 *	0.105	0.079	0.198 *	0.073	0.404 **	0.450 **
Asthma				0.404 **							
COPD ^b^		0.216 *	0.404 **				0.289 **			0.269 **	0.364 **
Stroke	0.217 *	0.215 *						0.270 **		0.451 **	0.285 **
Diabetes	0.240 **										0.208 *
Liver disease				0.289 **					0.243 **		
Kidney disease		0.198 *			0.270 **						0.263 **
Autoimmune disease							0.243 **				
ADL ^c^	0.581 **	0.404 **		0.269 **	0.451 **						0.796 **
Nutrition	0.450 **	0.450 **		0.364 **	0.285 **	0.208 *		0.263 **		0.796 **	

^a^ Heart disease in healthy individuals involves hypertension and, in frail elderly individuals, heart failure; ^b^ chronic obstructive pulmonary disease; ^c^ Activities of Daily Living. * < 0.0 s. ** *p* < 0.01.

**Table 2 healthcare-12-02393-t002:** Health conditions and drug treatment among 70 healthy adults divided into age groups and 52 frail elderly individuals, also presented as separate groups (20 nursing home residents (NHRs) and 32 patients undergoing hospital-affiliated home care (LAH)). Empty cells indicate that a condition/drug is not present.

	Healthy Individuals				Frail Elderly Patients		
Condition	All *n* = 70 (%)	27–49 Years *n* = 11 *n* (%)	50–65 Years *n* = 25 *n* (%)	66–80 Years *n* = 31 *n* (%)	NHRs/LAH *n* = 52 *n* (%)	NHRs *n* = 20 *n* (%)	LAH *n* = 32 *n* (%)
Dementia					16 (31) ***	14 (70)	2 (6)
Hypertension	28 (38)	3 (27)	8 (32)	17 (55)	0 (0) ***	0 (0)	0 (0)
Heart disease ^a^					43 (83) ***	18 (90)	25 (78)
Asthma	7 (10)		1 (4)	6 (19)	8 (15)	1 (1)	7 (22)
COPD ^b^					11 (21) ***	3 (15)	8 (25)
Stroke					7 (14) **	6 (30)	1 (3)
Diabetes	4 (6)		2 (8)	2 (6)	10 (19) **	4 (20)	6 (18)
Chronic liver disease					1 (2)	1 (5)	0 (0)
Chronic kidney disease					6 (12) **	1 (5)	5 (16)
Autoimmune disease	9 (13)	1 (9)	2 (8)	6 (19)	6 (12)	5 (25)	1 (3)
Thyroid disease	4 (6)	1 (9)		3 (10)	6 (12)	2 (10)	4 (13)
**Daily medication**							
Cortisone > 5 mg					15 (29) ***	1 (5)	14 (44)
Sedatives					29 (56) ***	7 (35)	22 (69)
Antidepressants					14 (27) ***	8 (40)	6 (19)
Paracetamol, NSAID ^c^	5 (7)	1 (9)		4 (13)	47 (90) ***	18 (90)	29 (91)
Blood thinning ^d^	1 (1)			1(3)	22 (42) ***	6 (30)	16 (50)
Risk for malnutrition					52 (100) ***	20 (100)	32 (100)
ADL ^e^ score	0 ± 0	0 ± 0	0 ± 0	0 ± 0	4 ± 2	7 ± 2	4 ± 3
Number of daily drugs	0.7 ± 0.8	0.4 ± 0.5	0.4 ± 0.7	1.0 ± 0.8	2.8 ± 1.4	2.3 ± 1.4	3.1 ± 1.3
Number of diagnoses	0.8 ± 0.8	0.5 ± 0.7	0.5 ± 0.9	1.1 ± 0.8	2.2 ± 1.3 ***	3.3 ± 1.4	1.8 ± 1.3

^a^ Heart disease, considered hypertension in healthy individuals; ^b^ chronic obstructive pulmonary disease; ^c^ non-steroidal anti-inflammatory; ^d^ Waran, ASA (acetylsalicylacid); ^e^ Activities of Daily Living. Comparison was made between healthy and frail individuals, with ** *p* < 0.01, and *** *p* < 0.001.

**Table 3 healthcare-12-02393-t003:** Variation in body temperature and oxygen saturation presented as means ± SD and minimum and maximum values, measured 5 mornings in a row, among 70 healthy and 52 frail elderly subjects.

	Healthy Individuals (*n* = 70)					Frail Elderly Patients (*n* = 52)				
	N	Min	Max	Mean	SD	N	Min	Max	Mean	SD
Body temperature °C										
Mean	69	34.6	37	36.1	0.47	51	35.0	37.4	36.6 ***	0.39
SD	69	0.05	0.83	0.28	0.16	51	0.05	0.73	0.28	0.15
Min value	69	34.3	36.7	35.7	0.51	51	34.5	37.2	36.2	0.47
Max value	69	34.8	37.7	36.5	0.49	51	35.4	37.7	36.9	0.40
Oxygen saturation %										
Mean	70	92	99	97	1.29	52	79	100	95 ***	3.3
SD	70	0	5	1.07	0.78	52	0	7	2.0	1.4
Min value	70	85	98	95	2.29	52	75	100	92	4.5
Max value	70	95	99	98	1.02	52	83	100	97	2.9

*** *p* < 0.001, when comparing mean values of body temperature and oxygen saturation.

**Table 4 healthcare-12-02393-t004:** The results of binary logistic regression analysis of conditions related to minimum body temperature of 36.2 °C or below and maximum body temperature of 36.3 °C or above in 70 healthy and 52 frail elderly individuals.

	B	S.E.	Wald	Df	Sig.	Exp (B)	95% CI EXP (B)	
							Lower	Upper
Max body temperature ≤ 36.2 °C								
Nutrition	−3.181	1.082	8.641	1	0.003	0.042	0.005	0.346
Heart disease ^a^	−1.833	0.938	3.815	1	0.051	0.160	0.025	1.006
ADL ^b^	−0.480	0.195	6.035	1	0.014	0.619	0.422	0.908
COPD ^c^ dementia, heart disease	−2.185	0.807	7.331	1	0.007	0.112	0.023	0.547
ADL ^b^, dementia, heart disease ^a^, nutrition	−1.384	0.508	7.423	1	0.006	0.250	0.093	0.678
Max body temperature ≥ 36.3 °C								
Nutrition	2.254	0.647	12.153	1	0.001	9.524	2.682	33.816
Heart disease ^a^	1.852	0.647	8.194	1	0.004	6.373	1.793	22.648
ADL ^b^	0.304	0.114	7.087	1	0.008	1.355	1.083	1.695
COPD ^c^, dementia, heart disease ^a^	1.125	0.419	7.223	1	0.007	3.081	1.356	6.999
ADL ^b^, dementia, nutrition, heart disease ^a^	0.820	0.264	9.636	1	0.002	2.270	1.353	3.809

^a^ Heart disease is considered hypertension in healthy subjects and heart failure in frail elderly participants; ^b^ Activities of Daily Living; ^c^ chronic obstructive pulmonary disease.

**Table 5 healthcare-12-02393-t005:** The results of binary logistic regression analysis of conditions related to minimum oxygen saturation of 94% or below and maximum oxygen saturation of 95% or above in 70 healthy and 52 frail individuals.

Condition	B	S.E.	Wald	df	Sig.	Exp (B)	95% CI Exp (B)	
							Lower	Upper
Min oxygen saturation ≤ 94%								
COPD	2.485	1.281	3.763	1	0.052	12.000	0.975	147.772
Asthma	2.817	1.260	4.999	1	0.025	16.734	1.416	197.814
Dementia	2.757	1.198	5.295	1	0.021	15.754	1.505	164.943
Diabetes	2.705	1.090	6.154	1	0.013	14.949	1.764	126.677
Max oxygen saturation ≥ 95%								
COPD ^a^	−3.005	0.858	12.277	1	0.001	0.050	0.009	0.266
Asthma	−3.466	1.163	8.882	1	0.003	0.031	0.003	0.305
Dementia	−2.531	1.258	4.049	1	0.044	0.080	0.007	0.936
Heart disease ^b^	−1.596	0.860	3.445	1	0.063	0.203	0.038	1.094

^a^ Chronic obstructive pulmonary disease. ^b^ Heart disease is considered hypertension in healthy subjects and heart failure in frail elderly patients.

**Table 6 healthcare-12-02393-t006:** Summary of the course, based on excerpts from health care records, when nursing home resident Asta, 91 years old, suffered from pneumonia.

**Day 1**	**Nursing record** Black eyebrow. She probably fell. She is worried, sorry, and cannot sleep. Tired and rested a lot today.
**Day 2**	**Nursing record** Morning ear temperature: 37.2 °C. Aggressive, anxious, confused, lethargic sad and crying, speech more incoherent than usual, unrestrained, infirm. Respiratory and UTI symptoms. She is not as usual. Nausea at dinner, went to bed. Moves with more stiffness.
**Day 3**	**Nursing record** Sitting on the floor in the morning. Sad, crying, stomach hurts. She is thoroughly tired.
**Day 4**	**Nursing record** She is very wheezy and has a temperature. Acute CRP ^a^; doctor orders antibiotics for pulmonary infection.
**Day 5**	**Nursing record** In the afternoon, respiratory distress. Called ambulance, obtained diuretic. Pulmonary edema (acute), so she was sent to hospital **Medical record** Subfebrile: 2–3 days, mucous cough, tired and infirm. CRP ^a^ > 100. Rectal temperature: 38.2 °C. Respiratory symptoms, for which oral antibiotics were provided.
**Day 6**	**Nursing record** Calling the hospital. Zinasef iv X-ray pulm. today.
**Day 12**	**Nursing record** Calling the hospital. Only received antibiotics intravenously in palliative care.

^a^ C-reactive protein.

## Data Availability

All available data related to this study are presented in the current manuscript.

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
