# Peer review of "The Development of Algorithms for Individual Ranges of Body Temperature and Oxygen Saturation in Healthy and Frail Individuals"

_healthcare, 2024, doi:10.3390/healthcare12232393_

Round 1

Reviewer 1 Report

Comments and Suggestions for Authors

Find the attached file .

The similarity is 45% it should be less than 10%. 

Author Response

Reviewer 1

The title is changed to The Development of Algorithms for individual ranges on body temperature and oxygen saturation in healthy and frail individuals

Thank you for valuable suggestions for improvement of our manuscript. We have revised accordingly. Line references refer to the document with all changes accepted and language review completed.

  1. Clarify Methodology: The description of the developed algorithms lacks detail. Briefly mentioning the algorithmic approach (e.g., statistical modelling, machine learning) and any key parameters would enhance clarity and allow readers to understand the underlying methods.
  2. Control Variables: Measurements were taken under controlled conditions, but expanding on the rationale behind these controls would help establish consistency and improve reproducibility.

Comments to 1 and 3: See especially Line 223 to 236.

We roughly describe the development of the algorithm, based on our earlier research, in the manuscript. The considerations we have taken into account in the development of the algorithm are:

The technical measurement uncertainty, i. e drift in the device, was estimated in accordance with the ISO standards (ISO. Medical electrical equipment_ Part 2-56: Particular requirements för basic safety and essential performance of clinical thermometers for body temperature measurement. ISO 80601-2-56: 2009:E. Switzerland: International Organization of Standardization; 2014; ISO. Medical electrical equipment – Part 2-61: Particular requirements for basic safety and essential performance of pulse oximeter equipment. ISO 80601-2-61:2017. International Organisation of Standardization; 2017).

The reproducibility of repeated measurements, including the 95% CI of the mean differences between the sets of measurements for the right ear. Inter-individual variability. We found no significant difference between repeated measurements performed by two operators (Sund-Levander M, Grodzinsky E, Loyd D, Wahren LK. Error in body temperature assessment related to individual variation, measuring technique and equipment. International Journal of Nursing Practice. 2004;10:216-23).

In a multicentre study, Difftemp® was defined as the difference between temperatures considered by the individual to be increased temperature in fever minus measured temperature, by age and sex in 1,706 men and women. Based on the results, instead of current cutoff values assessed as fever, we proposed that an increase of 1.0°C from the individual’s normal body temperature together with malaise, could be indicative of increased temperature defined as fever (Sund Levander M, Grodzinsky E. Variation in normal ear temperature. American Journal of the Medical Sciences. 2017;54(4):370-78.). This means that too large  deviation when establishing the individual normal range can indicate ongoing processes affecting body temperature. The measurements have to be postponed for 10 days and repeated. 

  1. Define Population Demographics: Further clarification on the criteria used to classify individuals as "healthy" or "frail" would strengthen the study. Including a detailed breakdown of age distribution and other demographics, especially regarding chronic illness and medication, would also enhance reliability.

Comment: We have added text to clarify classification of healthy and frail. Se Line 153 to 156.

Table 2 We have divided the healthy group also into subgroups and removed 0 (0) when the condition/drugs are not present, and hope this make the table more easy to read.

  1. Statistical Significance: While differences in body temperature and oxygen saturation between groups are highlighted, noting any statistical tests or significance levels would add depth and reinforce the validity of these observations.

Comment: We have added text in the Statistical section.to clarify. See Line 239 to 254.

  1. Explain Variation Measurements: The reported variations in temperature and oxygen saturation are unclear in terms of their calculation. Specifying whether these are standard deviations, ranges, or another metric would improve the reader's understanding.

Comment: We have added text in Table 3 and Figure 1 for clarification.

  1. Precision Medicine Relevance: The connection between individual habitual conditions and precision medicine is intriguing. Detailing how these findings could be applied in clinical settings, or providing examples where individual ranges might enhance patient care, would reinforce the practical value of the study.

Comment: After Discussion we have added a new section Clinical implications to more address practical implementation.  See Line 438 -469.

  1. Implications of Results: The suggestion that interpreting deviations from individual normal ranges could guide timely care is valuable. Including specific clinical scenarios or actions that could arise from these deviations would make the potential impact clearer.

Comment: In the Clinical Implications we have added a real-life scenario to illuminate clinical benefits.

  1. Sentence Correction Needed: There’s an incomplete sentence, "which can be d,” that should be completed or removed for professionalism and clarity.

Comment: Sorry, we cannot find such incomplete sentence.

  1. Acknowledge Limitations: No limitations are currently noted, which may leave readers questioning the study's robustness. Briefly acknowledging sample size constraints or variability in measurement methods would provide a more balanced perspective.

Comment: Text is added text about the limitations you observed in Limitations. See Line 476 to 483.

Reviewer 2 Report

Comments and Suggestions for Authors

Paper entitled “Applied Precision Medicine: Development of Algorithms that Calculates, Approves and Interprets Individual Reference Ranges for Body Temperature and Oxygen Saturation in Healthy and Frail Individuals” designed temperature and oxygen saturation assessments on the basis of Personalized Medicine procedure. It seems that the authors inspired the temperature and oxygen saturation directly from their 10-year clinical observations and also the pandemic COVID-19 and extended it to an employment of Personalized Medicine. However, the paper considered a Precision Medicine concept in their work, I feel this paper as two-edged sword both interesting and worrying. My major concern comes from the Precision and Personalized medicine themselves. These concepts are directly linked with the unique characteristics of a person or a group and thus, you can ought to include the Genetics and Genomics. Accordingly, the diverse genetic profiles of individuals lead to various clinical manifestations and personal responses to drugs and a distinct treatment. Due to some merits which can be found in this study, I suggest the reviewer to renew their title with a moderated and softened form about Precision Medicine and increase the focus of the title with new findings in Temperature and Oxygen Saturation compared to both classic standpoint and clinically practical utilities. Finally, here I listed a series of major and minor comments:

Major Comments:

1-      Plagiarism rate needs to be decreased [iThenticate found 45% of duplicity].

2-      They mentioned the role of Genetic just two times (one time in Introduction and one time in Discussion); however, Genetic is the heart of Personalized and Precision Medicine concept and should not be neglected. The authors should declare that they used a time- and cost- saving genetic-free solution for early filtration in reception of people for hospitalization vs. home-caring supervisions.

3-      Last paragraph of Results needs to be revised for the proper mathematical signs. See “When including medication in the logistic regression, painkillers and sedatives were negatively related with maximum body temperature 36.2°C or below (Exp B 0.250, 95% CI. 0.093 to 0.671, p 0.006, and Exp B 0.173, 95% CI. 0.038 to 0.779, p 0.022, respectively)” after “p” would it be = or < ?

4-      The authors should clarify about the limitations which their study is facing with. For instance, they should mention about the lack of investigations on each individual’s genetic markers involved in inflammation and metabolism.

Minor Comments:

The whole manuscript should be revised by an English Native Speaker for the improvement in understanding and readership of the items and also correcting the grammatical and typing errors occurred during the manuscript. Some of them are listed here for example:

1-      Bolded parts should be reconsider and remove the boldness; SEE> “Hence, we are not surprised that, as declared by Lohse [57], uncertainty seems to be a key characteristic of precision medicine in practice, in particular in relation to clinical decision‐making.”

2-      Colored words should be turned into black; SEE> This might explain the “higher mean” body temperature in frail elderly in the present study.

3-      Remove additional writing signs; SEE> “Assessing the health status of the frail elderly is a clinical challenge due to the complexity of ageing, frailty, and atypical presentation [16-21]..” Remove one dot after the bracket. Also, in “In the frail individuals, the risk of malnutrition and impaired physical and cognitive abilities, and daily medication with analgesics, sedatives and antidepressants. were related” correct the dot after antidepressants with a comma.

Comments on the Quality of English Language

An English Native Speaker needs to revise the whole manuscript for both comprehensiveness and grammatical aspects. 

Author Response

Reviewer 2

Comments and Suggestions for Authors

Paper entitled “Applied Precision Medicine: Development of Algorithms that Calculates, Approves and Interprets Individual Reference Ranges for Body Temperature and Oxygen Saturation in Healthy and Frail Individuals

designed temperature and oxygen saturation assessments on the basis of Personalized Medicine procedure. It seems that the authors inspired the temperature and oxygen saturation directly from their 10-year clinical observations and also the pandemic COVID-19 and extended it to an employment of Personalized Medicine. However, the paper considered a Precision Medicine concept in their work, I feel this paper as two-edged sword both interesting and worrying. My major concern comes from the Precision and Personalized medicine themselves. These concepts are directly linked with the unique characteristics of a person or a group and thus, you can ought to include the Genetics and Genomics. Accordingly, the diverse genetic profiles of individuals lead to various clinical manifestations and personal responses to drugs and a distinct treatment. Due to some merits which can be found in this study, I suggest the reviewer to renew their title with a moderated and softened form about Precision Medicine and increase the focus of the title with new findings in Temperature and Oxygen Saturation compared to both classic standpoint and clinically practical utilities. Finally, here I listed a series of major and minor comments:

Major Comments:

Comment: Title changed to The Development of Algorithms for individual ranges on body temperature and oxygen saturation in healthy and frail individuals

Thank you for valuable suggestions for improvement of our manuscript. We have revised accordingly. Line references refer to the document with all changes accepted and language review completed.

1-      Plagiarism rate needs to be decreased [iThenticate found 45% of duplicity].

Comment: We contacted the library Linköping University, who helped us with running the manuscript in the iThenticate. When the search engine included ESMED iThenticate found 45% of duplicity. This is explained as our first version of this manuscript was accepted by the predator journal Medical Research Archives, alleged and fake journal of the fake organization European Society of Medicine (ESMED). When we discovered that, we immediately withdrawn the paper from the journal, which is confirmed in email exchange with the Medical Research Archives journal. We also contacted Research web who confirmed withdrawl of the paper. We have informed the editors of Healtcare about this mistake, and they approved the manuscript as not before published. The librarian explained that iThenticate does not specify which period the search refers to, so the explanation for the 45% hit when ESMED is included in the search is likely to be that the search is within the period when the manuscript was handled by ESMED. In addition, the search in iThenticate is very broad. The important thing is that the manuscript has been removed, and is nowhere to be found on the web.

2-      They mentioned the role of Genetic just two times (one time in Introduction and one time in Discussion); however, Genetic is the heart of Personalized and Precision Medicine concept and should not be neglected. The authors should declare that they used a time- and cost- saving genetic-free solution for early filtration in reception of people for hospitalization vs. home-caring supervisions.

Comment: We added text in the method section Line 164 and your suggested text in the new section Clinical Implications: See Line 438 to 440.

3-      Last paragraph of Results needs to be revised for the proper mathematical signs. See “When including medication in the logistic regression, painkillers and sedatives were negatively related with maximum body temperature 36.2°C or below (Exp B 0.250, 95% CI. 0.093 to 0.671, p 0.006, and Exp B 0.173, 95% CI. 0.038 to 0.779, p 0.022, respectively)” after “p” would it be = or < ?

Comment: We have corrected the mistakes. See Line 332 to 339

4-      The authors should clarify about the limitations which their study is facing with. For instance, they should mention about the lack of investigations on each individual’s genetic markers involved in inflammation and metabolism.

Comment:  We did not include any genetic markers as we started from the individual's habitual state. We have added limitations of the study. See Line 476 to 483.

Minor Comments:

The whole manuscript should be revised by an English Native Speaker for the improvement in understanding and readership of the items and also correcting the grammatical and typing errors occurred during the manuscript. Some of them are listed here for example:

Comment: The manuscript is English edited by the MDPI service.

1-      Bolded parts should be reconsider and remove the boldness; SEE> “Hence, we are not surprised that, as declared by Lohse [57], uncertainty seems to be a key characteristic of precision medicine in practice, in particular in relation to clinical decision‐making.”

Comment: Sorry, we do not understand. We do not have any bold parts in the main text, other than headings.

2-      Colored words should be turned into black; SEE> This might explain the “higher mean” body temperature in frail elderly in the present study.

Comment: Sorry, we do not understand. We do not have any coloured words in the text

3-      Remove additional writing signs; SEE> “Assessing the health status of the frail elderly is a clinical challenge due to the complexity of ageing, frailty, and atypical presentation [16-21]..” Remove one dot after the bracket. Also, in “In the frail individuals, the risk of malnutrition and impaired physical and cognitive abilities, and daily medication with analgesics, sedatives and antidepressants. were related” correct the dot after antidepressants with a comma.

Comment: Changes made accordingly. See Line 350 to 354.

Reviewer 3 Report

Comments and Suggestions for Authors

Thank you for allowing me the opportunity to review the article titled "Applied Precision Medicine: Development of Algorithms that Calculate, Approve, and Interpret Individual Reference Ranges for Body Temperature and Oxygen Saturation in Healthy and Frail Individuals." The authors present an innovative approach to precision medicine by developing algorithms for individualized reference ranges for vital parameters. The study holds potential for significantly enhancing clinical decision-making, especially in frail elderly populations. However, several concerns arise throughout the manuscript.

Firstly, the title is way too long - make it more crisp and attractive. Other major concerns are as follows:

1. Narrative Flow and Grammar:

Strength: The writing is clear and technical, effectively communicating the study's purpose and methodology.

Concern: The use of passive voice throughout the article hinders readability and engagement. Active voice would improve the flow and make the findings more accessible. For instance, the results section could use active voice to describe the relationships found (e.g., "We observed that frail elderly individuals showed larger variations in oxygen saturation").

Recommendation: Simplify long sentences and make the writing more direct using active voice to enhance clarity and reader engagement.

2. Novelty and Relevance:

The study presents a novel contribution by establishing individual normal ranges for body temperature and oxygen saturation, especially focusing on frail populations. The concept of using algorithms to interpret deviations from individual norms is timely and essential in the context of personalized medicine.

Major Concern: While the study’s focus is on precision medicine, it does not fully address the practical implementation of these algorithms in clinical settings. A deeper discussion on how these algorithms will integrate with existing healthcare technologies would add value.

3. Methodology:

The descriptive design and the use of two distinct populations (healthy vs frail) are appropriate. However, the sample size for frail individuals (52) is relatively small and limits the generalizability of the findings.

Missing details: The methodology lacks detailed information on the validation of the algorithms. What specific thresholds were used for validation, and how were these validated against gold-standard clinical practices?

Recommendation: A more rigorous explanation of algorithm development and validation is required, including whether any sensitivity analyses were conducted to ensure robustness.

4. Statistical Tools Employed:

The statistical analysis, particularly the use of binary logistic regression, is sound. However, the authors do not provide sufficient justification for some of their dichotomous cut-offs, such as defining low body temperature as ≤36.2°C and low oxygen saturation as ≤94%. A more detailed explanation of how these thresholds were chosen is necessary.

5. Results:

The results provide a detailed breakdown of body temperature and oxygen saturation variations. However, the tables are dense and could benefit from additional summary statistics to highlight the most significant findings. For instance, Table 2 provides an overwhelming amount of data without emphasizing key differences between groups.

Recommendation: Simplify tables by summarizing key trends, such as highlighting the significance of variation in frail elderly individuals compared to healthy individuals.

6. Discussion:

The discussion appropriately places the findings within the context of precision medicine, particularly highlighting the large variations in body temperature and oxygen saturation in frail populations. The link between frailty, chronic illness, and vital signs is well-made.

Major Concern: The discussion lacks a comparative analysis with other studies that have examined individualized reference ranges or personalized algorithms. Including comparisons with recent studies on precision medicine in the elderly or frail populations would strengthen the argument.

Recommendation: Incorporate more recent research to show how this study advances the field. For example, research on variability in oxygen saturation in elderly populations during the COVID-19 pandemic could provide valuable context.

7. Limitations:

The authors briefly mention that healthy individuals were not confirmed with medical records. This introduces potential bias as some participants may have undiagnosed conditions.

Recommendation: A more thorough discussion of study limitations is needed, including potential biases in data collection, the small sample size for frail individuals, and the lack of external validation for the algorithms.

8. Conclusion:

The conclusion rightly emphasizes the potential for these algorithms to enhance precision medicine. However, the clinical applicability of these findings could be further explored. How do the authors envision these algorithms being used in routine practice? Will they be integrated into existing electronic health records or used as standalone tools?

Recommendation: Provide more concrete suggestions on the next steps for clinical implementation and further validation of the algorithms in larger, more diverse populations.

Comments on the Quality of English Language

Grammar is fine but it doesn't flow well.

Author Response

Reviewer 3

Comments and Suggestions for Authors

Thank you for allowing me the opportunity to review the article titled "Applied Precision Medicine: Development of Algorithms that Calculate, Approve, and Interpret Individual Reference Ranges for Body Temperature and Oxygen Saturation in Healthy and Frail Individuals." The authors present an innovative approach to precision medicine by developing algorithms for individualized reference ranges for vital parameters. The study holds potential for significantly enhancing clinical decision-making, especially in frail elderly populations. However, several concerns arise throughout the manuscript.

Firstly, the title is way too long - make it more crisp and attractive. Other major concerns are as follows:

Comment: Title changed to The Development of Algorithms for individual ranges on body temperature and oxygen saturation in healthy and frail individuals

Thank you for valuable suggestions for improvement of our manuscript. We have revised accordingly. Line references refer to the document with all changes accepted and language review completed.

  1. Narrative Flow and Grammar:

Strength: The writing is clear and technical, effectively communicating the study's purpose and methodology.

Concern: The use of passive voice throughout the article hinders readability and engagement. Active voice would improve the flow and make the findings more accessible. For instance, the results section could use active voice to describe the relationships found (e.g., "We observed that frail elderly individuals showed larger variations in oxygen saturation").

Recommendation: Simplify long sentences and make the writing more direct using active voice to enhance clarity and reader engagement.

Comment: We have generalized revised the text and has undergone English editing by it is the MDPI service.

  1. Novelty and Relevance:

The study presents a novel contribution by establishing individual normal ranges for body temperature and oxygen saturation, especially focusing on frail populations. The concept of using algorithms to interpret deviations from individual norms is timely and essential in the context of personalized medicine.

Major Concern: While the study’s focus is on precision medicine, it does not fully address the practical implementation of these algorithms in clinical settings. A deeper discussion on how these algorithms will integrate with existing healthcare technologies would add value.

Comment: After Discussion we have added a new section Clinical implications to more address practical implementation.  See Line 438 to 469.

  1. Methodology:

The descriptive design and the use of two distinct populations (healthy vs frail) are appropriate. However, the sample size for frail individuals (52) is relatively small and limits the generalizability of the findings.

Missing details: The methodology lacks detailed information on the validation of the algorithms. What specific thresholds were used for validation, and how were these validated against gold-standard clinical practices?

Recommendation: A more rigorous explanation of algorithm development and validation is required, including whether any sensitivity analyses were conducted to ensure robustness.

Comments: See especially Line 223 to 236. We roughly describe the development of the algorithm, based on our earlier research, in the manuscript. The considerations we have taken into account in the development of the algorithm are:

The technical measurement uncertainty, i. e drift in the device, was estimated in accordance with the ISO standards (ISO. Medical electrical equipment_ Part 2-56: Particular requirements för basic safety and essential performance of clinical thermometers for body temperature measurement. ISO 80601-2-56: 2009:E. Switzerland: International Organization of Standardization; 2014; ISO. Medical electrical equipment – Part 2-61: Particular requirements for basic safety and essential performance of pulse oximeter equipment. ISO 80601-2-61:2017. International Organisation of Standardization; 2017).

The reproducibility of repeated measurements, including the 95% CI of the mean differences between the sets of measurements for the right ear. Inter-individual variability. We found no significant difference between repeated measurements performed by two operators (Sund-Levander M, Grodzinsky E, Loyd D, Wahren LK. Error in body temperature assessment related to individual variation, measuring technique and equipment. International Journal of Nursing Practice. 2004;10:216-23).

In a multicentre study, Difftemp® was defined as the difference between temperatures considered by the individual to be increased temperature in fever minus measured temperature, by age and sex in 1,706 men and women. Based on the results, instead of current cutoff values assessed as fever, we proposed that an increase of 1.0°C from the individual’s normal body temperature together with malaise, could be indicative of increased temperature defined as fever (Sund Levander M, Grodzinsky E. Variation in normal ear temperature. American Journal of the Medical Sciences. 2017;54(4):370-78.). This means that too large deviation when establishing the individual normal range can indicate ongoing processes affecting body temperature. The measurements have to be postponed for 10 days and repeated. 

The algorithms has yet to be validated clinically, including sensitivity analysis.

  1. Statistical Tools Employed:

The statistical analysis, particularly the use of binary logistic regression, is sound. However, the authors do not provide sufficient justification for some of their dichotomous cut-offs, such as defining low body temperature as ≤36.2°C and low oxygen saturation as ≤94%. A more detailed explanation of how these thresholds were chosen is necessary.

Comment: As there is no consensus of low limit for normal body temperature or oxygen saturation we used our own research and WHO to guide us. In Statistics, the text is revised to

Clarify. See Line 239 to 254.

  1. Results:

The results provide a detailed breakdown of body temperature and oxygen saturation variations. However, the tables are dense and could benefit from additional summary statistics to highlight the most significant findings. For instance, Table 2 provides an overwhelming amount of data without emphasizing key differences between groups.

Recommendation: Simplify tables by summarizing key trends, such as highlighting the significance of variation in frail elderly individuals compared to healthy individuals.

Comment: Table 1 is now presenting only significant correlations

Table 2 We have divided the healthy group also into subgroups and removed 0 (0) when the condition/drugs are not present, and hope this make the table more easy to read.

Figure 1. We have added explaining text about the boxplots.

  1. Discussion:

The discussion appropriately places the findings within the context of precision medicine, particularly highlighting the large variations in body temperature and oxygen saturation in frail populations. The link between frailty, chronic illness, and vital signs is well-made.

Major Concern: The discussion lacks a comparative analysis with other studies that have examined individualized reference ranges or personalized algorithms. Including comparisons with recent studies on precision medicine in the elderly or frail populations would strengthen the argument.

Recommendation: Incorporate more recent research to show how this study advances the field. For example, research on variability in oxygen saturation in elderly populations during the COVID-19 pandemic could provide valuable context.

Comment. A major problem is that we cannot find publications that examine individualised reference ranges. We highlighted this in the beginning of the Discussion. See Line 375 to 379.

Current reference ranges for vital parameters, including body temperature and oxygen saturation, are based on mean values on a group level. Also, people with long-term medical conditions and with daily medication are excluded from reference populations. Current interpretation of results entails a risk of misinterpretation on each measurement occasion because a deviant value for the individual may be camouflaged by a normal value at group level. We have included 2 new references, Möckel et. al and and Lapun et.al. about low oxygen saturation in COVID and elderly. See Line 85 to 88 and Line 111 to 112.

  1. Limitations:

The authors briefly mention that healthy individuals were not confirmed with medical records. This introduces potential bias as some participants may have undiagnosed conditions.

Recommendation: A more thorough discussion of study limitations is needed, including potential biases in data collection, the small sample size for frail individuals, and the lack of external validation for the algorithms.

Comment: Text is added text about the limitations you observed in Limitations. See Line 476 to 483.

  1. Conclusion:

The conclusion rightly emphasizes the potential for these algorithms to enhance precision medicine. However, the clinical applicability of these findings could be further explored. How do the authors envision these algorithms being used in routine practice? Will they be integrated into existing electronic health records or used as standalone tools?

Recommendation: Provide more concrete suggestions on the next steps for clinical implementation and further validation of the algorithms in larger, more diverse populations.

Comment: We have added a section Clinical implications to more address concrete suggestion. See Line 476 to 483.

Thank you for reading our paper and for valuable comments!

Reviewer 4 Report

Comments and Suggestions for Authors

Thank you for the opportunity to review the manuscript titled:  Applied Precision Medicine: Development of Algorithms that Calculates, Approves and Interprets Individual Reference Ranges for Body Temperature and Oxygen Saturation in Healthy and Frail Individuals

The study presents an important innovation in precision medicine by developing algorithms for individualized reference ranges for body temperature and oxygen saturation. These algorithms enhance clinical decision-making by considering personal baselines rather than universal thresholds, which may not accurately reflect variations among healthy and frail individuals. This approach addresses the limitations of standard reference values in detecting subtle changes in frail elderly patients who often display non-standard symptoms of illness. The work in my opinion has several limitations that need to be acknowledged in the discussion/limitations section.

My comments to the authors are the following:

1. The introduction section could be shorter and denser in meaning.

2. The study includes 75 participants, a modest sample that might limit the generalizability of the findings, especially given the variability among elderly populations. Expanding the sample size or including additional sub-groups would strengthen the conclusions.

3. While the study outlines the algorithm's development, further clinical validation of DiffTemp® and DiffOx® in a larger clinical setting would be essential. Including details on a potential validation protocol, with specific performance metrics, would enhance confidence in the applicability of these algorithms in clinical practice.

4. Medication types and dosages were recorded, but other lifestyle factors (e.g., diet, sleep quality, activity levels) could impact body temperature and oxygen saturation and may warrant additional control or discussion.

5. Although the study mentions algorithm development, it lacks a comprehensive explanation of the algorithmic process, thresholds, or decision rules used to determine individualized reference ranges. This addition would allow for replication in future studies.

6. The study would benefit from a more detailed exploration of how these individualized algorithms could be integrated into clinical workflows. 

7. The length of the discussion section could be shortened by keeping the focus on the study findings and their implications. For example, in my opinion, the first two paragraphs of the discussion do not add significantly to the manuscript and could be omitted. 

Author Response

Reviewer 4

Comment: Title changed to The Development of Algorithms for individual ranges on body temperature and oxygen saturation in healthy and frail individuals

Thank you for valuable suggestions for improvement of our manuscript. We have revised accordingly. Line references refer to the document with all changes accepted and language review completed.

Comments and Suggestions for Authors

Thank you for the opportunity to review the manuscript titled:  Applied Precision Medicine: Development of Algorithms that Calculates, Approves and Interprets Individual Reference Ranges for Body Temperature and Oxygen Saturation in Healthy and Frail Individuals

The study presents an important innovation in precision medicine by developing algorithms for individualized reference ranges for body temperature and oxygen saturation. These algorithms enhance clinical decision-making by considering personal baselines rather than universal thresholds, which may not accurately reflect variations among healthy and frail individuals. This approach addresses the limitations of standard reference values in detecting subtle changes in frail elderly patients who often display non-standard symptoms of illness. The work in my opinion has several limitations that need to be acknowledged in the discussion/limitations section.

My comments to the authors are the following:

  1. The introduction section could be shorter and denser in meaning.

Comment: We have shortened the introduction. In some way, but found the remaining text relevant.

  1. The study includes 75 participants, a modest sample that might limit the generalizability of the findings, especially given the variability among elderly populations. Expanding the sample size or including additional sub-groups would strengthen the conclusions.

Comment: The sample includes 122 subjects, 70 healthy and 52 frail individuals. See Line 260 to 264.

  1. While the study outlines the algorithm's development, further clinical validation of DiffTemp® and DiffOx® in a larger clinical setting would be essential. Including details on a potential validation protocol, with specific performance metrics, would enhance confidence in the applicability of these algorithms in clinical practice.

Comment: We are planning further clinical validation studies of of DiffTemp® and DiffOx®, and specifically of the algorithms, including sensitivity analysis.

  1. Medication types and dosages were recorded, but other lifestyle factors (e.g., diet, sleep quality, activity levels) could impact body temperature and oxygen saturation and may warrant additional control or discussion.

Comment: Thank you for notifying this. We did not address these issues in this study. They can be the focus in upcoming studies.

  1. Although the study mentions algorithm development, it lacks a comprehensive explanation of the algorithmic process, thresholds, or decision rules used to determine individualized reference ranges. This addition would allow for replication in future studies.

Comments: See especially Line 223 to 236.

We roughly describe the development of the algorithm, based on our earlier research, in the manuscript. The considerations we have taken into account in the development of the algorithm are:

The technical measurement uncertainty, i. e drift in the device, was estimated in accordance with the ISO standards (ISO. Medical electrical equipment_ Part 2-56: Particular requirements för basic safety and essential performance of clinical thermometers for body temperature measurement. ISO 80601-2-56: 2009:E. Switzerland: International Organization of Standardization; 2014; ISO. Medical electrical equipment – Part 2-61: Particular requirements for basic safety and essential performance of pulse oximeter equipment. ISO 80601-2-61:2017. International Organisation of Standardization; 2017).

The reproducibility of repeated measurements, including the 95% CI of the mean differences between the sets of measurements for the right ear. Inter-individual variability. We found no significant difference between repeated measurements performed by two operators (Sund-Levander M, Grodzinsky E, Loyd D, Wahren LK. Error in body temperature assessment related to individual variation, measuring technique and equipment. International Journal of Nursing Practice. 2004;10:216-23).

In a multicentre study, Difftemp® was defined as the difference between temperatures considered by the individual to be increased temperature in fever minus measured temperature, by age and sex in 1,706 men and women. Based on the results, instead of current cutoff values assessed as fever, we proposed that an increase of 1.0°C from the individual’s normal body temperature together with malaise, could be indicative of increased temperature defined as fever (Sund Levander M, Grodzinsky E. Variation in normal ear temperature. American Journal of the Medical Sciences. 2017;54(4):370-78.). This means that too large  deviation when establishing the individual normal range can indicate ongoing processes affecting body temperature. The measurements have to be postponed for 10 days and repeated. 

  1. The study would benefit from a more detailed exploration of how these individualized algorithms could be integrated into clinical workflows. 

Comment: After Discussion we have added a new section Clinical implications to more address practical implementation.  See Line 438 to 469.

  1. The length of the discussion section could be shortened by keeping the focus on the study findings and their implications. For example, in my opinion, the first two paragraphs of the discussion do not add significantly to the manuscript and could be omitted. 

Comment: We think it important to put our approach in the context of prevailing paradigm, using mean values on group level when assessing the individual health status. But, we have concentrated on our results in the most part of the discussion and moved the text om normality etc to th

Round 2

Reviewer 2 Report

Comments and Suggestions for Authors

The authors addressed all of the comments and there is no further comment; thus, the revised version of the paper can be considered for acceptance. 

All the Best

Author Response

Dear reviewer

Thank you! 

All the best

Märta and Ewa

Reviewer 3 Report

Comments and Suggestions for Authors

Way better now, but still needs work on grammar and flow. The sample size has not been defined correctly yet. Descriptive study is a very broad term.  Its better if study design, study setting, and sampling is explained in more detail. 

Comments on the Quality of English Language

Moderate. Still not the best. 

Author Response

Referee 3 Round 2. 2024-11-24

Way better now, but still needs work on grammar and flow. Comments on the Quality of English Language; Moderate. Still not the best. 

Comment: We have changed/removed sentences to shorten the text. The manuscript has undergone English language editing by MDPI. Since the manuscript has been language checked according to the recommendation of the journal, we kindly ask the editor to decide whether the grammar etc is now OK. See certificate below.

“We certify that the following article

Applied Precision Medicine: Development of Algorithms that Calculates, Approves and

Interprets Individual Reference Ranges for Body Temperature and Oxygen Saturation in

Healthy and Frail Individuals

Märta Sund Levander *, Ewa Grodzinsky

has undergone English language editing by MDPI. The text has been checked for correct use of grammar and common

technical terms, and edited to a level suitable for reporting research in a scholarly journal.

MDPI uses experienced, native English speaking editors. Full details of the editing service can be found at

► https://www.mdpi.com/authors/english.

Basel, Switzerland

November 2024 english-86911”

The sample size has not been defined correctly yet. Descriptive study is a very broad term.  Its better if study design, study setting, and sampling is explained in more detail. 

Comment: We have added text in Methods to clarify the design, setting and sampling.

  1. Materials and methods- Line 160-161: This study was a cohort study with a descriptive and prospective design.

2.1.1. Healthy subjects: Line 170-173 .Healthy people managed their daily lives independently and without the need for any support from society, lived in their own accommodation and worked professionally or were retired.

2.1.2. Frail subjects. Line 183-188. Frail elderly people lived in three non-profit municipality nursing homes with support to cope with daily life around the clock. Patients who underwent LAH were cared for either in their own home with round-the-clock care, or in a special hospital ward. People in terminal care were not invited to participate.

Results

In the Results we have added information about the presence of no or one diagnosis in healthy subjects: Line 316-317: In healthy subjects, the majority (59/70; 84%) had no (n= 33) or one (n= 26)

Limitations We have added text Line 522-523: subjects, and not documented life style factors, e.g. sleep habits. Daily medication, though,

Conclusion. We have improved the conclusion Line 538-542.

Reviewer 4 Report

Comments and Suggestions for Authors

Thank you for the opportunity to review the revised manuscript. The authors have made several changes, and some of the points raised in my previous report have been addressed. In my view, the manuscript could benefit from further refinement, particularly by reducing its length. Additionally, any points noted in the previous report that cannot be fully addressed in this version might be included in a limitations section.

Author Response

Referee 4 Round 2

Thank you for the opportunity to review the revised manuscript. The authors have made several changes, and some of the points raised in my previous report have been addressed. In my view, the manuscript could benefit from further refinement, particularly by reducing its length. Additionally, any points noted in the previous report that cannot be fully addressed in this version might be included in a limitations section.

Way better now, but still needs work on grammar and flow. Comments on the Quality of English Language; Moderate. Still not the best. 

Comment: We have changed/removed sentences to shorten the text. The manuscript has undergone English language editing by MDPI. Since the manuscript has been language checked according to the recommendation of the journal, we kindly ask the editor to decide whether the grammar etc is now OK. See certificate below.

“We certify that the following article

Applied Precision Medicine: Development of Algorithms that Calculates, Approves and

Interprets Individual Reference Ranges for Body Temperature and Oxygen Saturation in

Healthy and Frail Individuals

Märta Sund Levander *, Ewa Grodzinsky

has undergone English language editing by MDPI. The text has been checked for correct use of grammar and common

technical terms, and edited to a level suitable for reporting research in a scholarly journal.

MDPI uses experienced, native English speaking editors. Full details of the editing service can be found at

► https://www.mdpi.com/authors/english.

Basel, Switzerland

November 2024 english-86911”

The sample size has not been defined correctly yet. Descriptive study is a very broad term.  Its better if study design, study setting, and sampling is explained in more detail. 

Comment: We have added text in Methods to clarify the design, setting and sampling.

  1. Materials and methods- Line 160-161: This study was a cohort study with a descriptive and prospective design.

2.1.1. Healthy subjects: Line 170-173 .Healthy people managed their daily lives independently and without the need for any support from society, lived in their own accommodation and worked professionally or were retired.

2.1.2. Frail subjects. Line 183-188. Frail elderly people lived in three non-profit municipality nursing homes with support to cope with daily life around the clock. Patients who underwent LAH were cared for either in their own home with round-the-clock care, or in a special hospital ward. People in terminal care were not invited to participate.

Results

In the Results we have added information about the presence of no or one diagnosis in healthy subjects: Line 316-317: In healthy subjects, the majority (59/70; 84%) had no (n= 33) or one (n= 26)